# SUCCINCT COMPRESSION: NEAR-OPTIMAL AND LOSSLESS COMPRESSION OF DEEP NEURAL NETWORKS DURING INFERENCE RUNTIME

## ABSTRACT

Recent advances in Deep Neural Networks (DNN) compression (e.g. pruning, quantization and etc.) significantly reduces the amount of space consumption for storage, making them easier to deploy in low-cost devices. However, those techniques do not keep the compressed representation during inference runtime, which incurs significant overheads in terms of both performance and space consumption. We introduce "Succinct Compression", a three-stage framework to enable DNN inference with near-optimal compression and much better performance during inference runtime. The key insight of our method leverages the concept of *Succinct Data Structures*, which supports fast queries directly on compressed representation without decompression. Our method first transforms DNN models as our proposed formulations in either Element-wise or Block-wise manner, so that *Succinct Data Structures* can take advantage of. Then, our method compresses transformed DNN models using *Succinct Data Structures*. Finally, our method exploits our specialized execution pipelines for different model formulations, to retrieve relevant data for DNN inference. Our experimental results show that, our method keeps near-optimal compression, and achieves at least 8.7X/11.5X speedup on AlexNet/VGG-16 inference, compared with Huffman Coding. We also experimentally show that our method is quite synergistic with Pruning and Quantization.

## 1 INTRODUCTION

Deep neural networks (DNNs) demand an increasing number of parameters as the required complexity of tasks, which substantially make DNN models become larger Bengio et al. (2021). Therefore, DNNs incurs a significant amount of memory footprints during inference runtime, and thus affects both the overall performance and the space consumption. Recent efforts compress DNN models in both loss (e.g. pruning, quantization and etc.) and lossless manners (e.g. Huffman Coding). However, the emphasis of the prior works focus on compressing DNN models for efficient storage, rather than providing space-efficient representations during the inference runtime. Thus, prior approaches require to decompress compressed model first and perform the inference, which consumes a huge amount of memory space. An alternative is to query the compressed models, decode the query results and perform the inference. However, this method suffers from significant performance overheads during the inference runtime.

Our goal is to improve the inference performance, while keeping the model compressed near-optimally. We introduce "Succinct Compression", a three-stage framework to enable much faster DNN inference with near-optimal and lossless compression simultaneously. The unique characteristic of our method is the fast-queryable yet near-optimally-compressed data structures called *Succinct Data Structures*. By engineering the inner operators, *Succinct Data Structures* allow fast lookup within the compressed representations directly, without decompressing them first. To exploit this unique set of data structures, we introduce two additional stages to make them and DNN models more synergistic: (1) before compressing DNN models into *Succinct Data Structures*, we propose two semi-structured formulations to represent DNN models in element-wise or block-wise manners; and (2) after maintaining DNN models in *Succinct Data Structures*, we specialize two pipelines for different formulations, by carefully engineering the inner operators of *Succinct Data Structures*.

The contributions of this paper are as follows.

(i) To improve the inference performance while keeping DNN model compressed, we introduce "Succinct Compression": a three-stage framework to exploit *Succinct Data Structures* to enable faster DNN inference with near-optimal compression at the same time.

(ii) We suggest two semi-structured formulations to store DNN models, which *Succinct Data Structures* can exploit them more efficiently. One formulates the DNN models in Element-wise manner, while the other one formulates the models in Block-wise manner.

(iii) We provide two approaches for the inference of compressed DNN models, maintained in *Succinct Data Structures*, using its inner operators. One serves for the inference pipeline in Element-wise manner, while the other serves for the inference pipeline in Block-wise manner.

(iv) We experimentally demonstrate that our method achieve significant speedup with near-optimal compression, compared with the state-of-the-art approach (i.e. Huffman Coding). In addition, our experiments also justify that our method is well-synergistic with other compression schemes (i.e. Pruning and Quantization).

## 2 RELATED WORKS

The architectures of DNN models grow much larger due to its incredible effects for resolving non-linear tasks. However, the rapidly growing size of DNN models incurs significant overheads in terms of storage. We first classify and elaborate modern compression mechanisms for DNN models into three parts: Pruning, Quantization and Model Encoding. Then we identify the novelty of our method, by comparing it with the above state-of-the-art approaches.

**Pruning** refers to those techniques enabling the removal of irrelevant units (weights, neurons or convolutional filters) (LeCun et al. (1989)). Relevance of weights is often determined by the absolute value ("magnitude based pruning" (Han et al. (2016; 2017); Guo et al. (2016)), but more sophisticated methods have been known for decades, e.g., based on second-order derivatives (Optimal Brain Damage (LeCun et al. (1989)) and Optimal Brain Surgeon (Hassibi & Stork (1992)) or ARD (automatic relevance determination, a Bayesian framework for determining the relevance of weights, (Neal (1996); Mackay & David (1995); Karaletsos & Rätsch (2015))).

**Quantization** refers to those techniques aimming for the reduction of the bit-precision of weights, activations or even gradients, which is highly demanded for hardware accelerator designs (Sze et al. (2017)). Methods range from fixed bit-width computation (e.g., 12-bit fixed point) to aggressive quantization such as binarization of weights and activations (Courbariaux et al. (2016); Rastegari et al. (2016); Zhou et al. (2016); Hubara et al. (2017)). Few-bit quantization (2 to 6 bits) is often performed by k-means clustering of trained weights with subsequent fine-tuning of the cluster centers (Han et al. (2016)). Pruning and quantization methods have been shown to work well in conjunction (Han et al. (2016)). In so-called "ternary" networks, weights can have one out of three possible values (negative, zero or positive) which also allows for simultaneous pruning and few-bit quantization (Li & Liu (2016); Zhu et al. (2017)).

**Model Encoding** refers to exploiting existing compression techniques to improve the space efficiency of model storage. These techniques are usually lossless, and compress DNN models via extra encoding. Han et al. (2015) leverages Huffman Encoding to reduce the storage of pruned and quantized DNN models by tens of magnitude. It uses variable-length codewords to encode source symbols. The table is derived from the occurrence probability for each symbol. More common symbols are represented with fewer bits. Huffman Coding is the optimal scheme for lossless compression, and Han et al. (2015) shows that it's synergistic with pruning and quantization. Our method aims to provide a more efficient mechanism over Huffman Coding, to optimize the inference performance while keeping DNN models compressed.

**Novelty of our method** comes from the following three aspects, compared with the above three methods. First, prior works from Model Encoding focuses on offline storage of DNN models, but our method targets to keep the compression and accelerate the inference during runtime; second, prior works, such as Pruning and Quantization, may cause loss of information, but our method is lossless and near-optimal; and third, prior works from Pruning and Quantization are not contradictory with our method, instead we show and demonstrate that they are synergistic with our method.

## 3 Formulating DNN Models

The first stage of our method is to form DNN models appropriately, so that *Succinct Data Structures* can take advantage of. *Succinct Data Structures* exploits the delimiters within a long string, to perform fast queries directly on the compressed string. Hence, our method first formulates DNN models as Runtime-Accessible Sequence (RAS), which refers to semi-structured format using a minimal amount of delimiters to construct hierarchical information (e.g. layers). To exploit *Succinct Data Structures* effectively, we propose two semi-structured formulations to store model details. First, we introduce Element-wise RAS by using delimiters to separate different elementary operands within DNN models; and second, we introduce Block-wise RAS by applying delimiters to separate different sets of data operands within DNN models, based on the related computation kernels. Based on the above Element-wise and Block-wise RAS, we provide specializations to make them synergistic with Pruning and Quantization, which can further improves both the performance and compression rate during inference runtime.

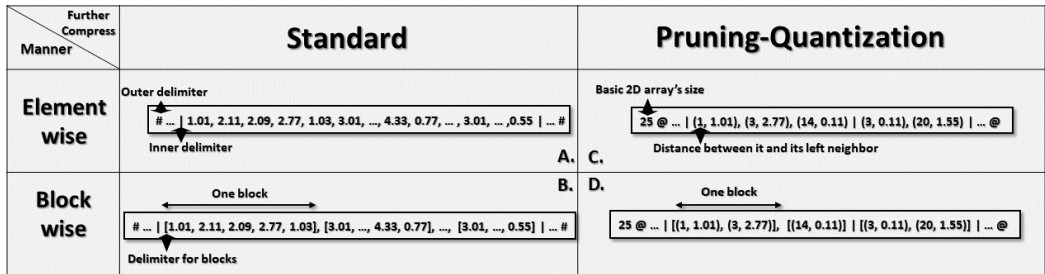

Figure 1: A comparison of different kinds of Runtime-Accessible Sequence (RAS).

### 3.1 Element-wise Runtime-Accessible Sequence

One type of formulation, suggested by our method, is Element-wise Runtime Accessible Sequence (denoted as Element-wise RAS). Element-wise RAS utilizes delimiters to separate elementary data operands. In the context of DNN models, the pre-defined delimiters (e.g. vertical bar and number sign) are used at the boundaries of different elementary data operands from DNN models, and these delimiters are used to query for elementary data operands accordingly.

Figure 1-(A) shows an example of Element-wise RAS; there are two vertical bars encompass several elementary operands. This methodology forms the Element-wise RAS, and the number sign is used to represent the border of this union. To properly formulate the whole network into Element-wise RAS, concatenate such unions by using a separate delimiter (e.g. '#').

### 3.2 Block-wise Runtime-Accessible Sequence

One limitation of Element-wise RAS is that frequent queries are required for every single data operand, before the computation for model inference. Therefore, to improve the efficiency of operand query, we suggest the other formulation of DNN models: Block-wise Runtime-Accessible Sequence (denoted as Block-wise RAS). Different from Element-wise RAS, Block-wise RAS forms basic building blocks for query and access based on the computation kernels, namely denoted as a block. Such a block stores a consecutive number of elementary data operands, which are used for a computation kernel. Between different blocks, Block-wise RAS exploits delimiters for separation, so that they can be efficiently queried.

Figure 1-(B) shows an example of Block-wise RAS: the Block-wise RAS aggregates five operands with two square brackets, as one individual block. This transformation of elementary operands, by synthesizing multiple operands and using a distinct delimiters, can provide faster queries by extracting them at one time, compared with Element-wise RAS.

### 3.3 Pruning-Quantization RAS

The designs of Element/Block-wise RAS are still without the consideration of Pruning and Quantization. As revealed by prior works, the impacts of these compression schemes can substantially incur a huge amount of sparsity within the model storage. Therefore, the designs of RAS need to account for this feature when the models are pruned and/or quantized. We provide generic optimization to make RAS synergistic with Pruning and Quantization, regardless of Element- or Block-wise. The key insight is to form elementary data operand in a similar manner as inverted indexes, by forming a tuple consisting of the exact values and the relative positions. Figure 1-(C)/(D) shows examples of the optimized Element-/Block-wise RAS for Pruning and Quantization.

Figure 1-(C) and (D) shows examples of Pruning-Quantization RAS for Element-wise and Block-wise. The difference hereby is that, we refine the elementary operands as tuples. In such a tuple, the first element refers to the relative distance between this and its left neighbor (a number or a delimiter); and the second element stores the value of the corresponding data operand. This approach is synergistic with Pruning and Quantization because: (1) for Pruning, the relative distance, contained in the reshaped tuple, can effectively exploit the pruned model structures; and (2) for Quantization, the values, contained in the reshaped tuple, can enhance the space benefits from quantization.

## 4 Succinct Data Structures

After formulating DNN models into RAS, the second stage of our method stores them in *Succinct Data Structures* during runtime. *Succinct Data Structures* were first pioneered by Jacobson (1988), which refers to a set of data structures using the near-information-theoretic bound space to store the compressed representation, and still provide fast query and access operations directly on these compressed representations. In general, *Succinct Data Structures* have the following representative inner operators.

Given a string $S$ whose length and alphabet are $L$ and $\sigma$, there are three operations directly on the compression (shown below).

- $Rank_q(x)$ returns the number of symbol $q$ appearing in $S_{0:x}$ where $q \in \sigma$ and $x < L$.

- $Select_q(x)$ returns the position of $x$-th occurrence of symbol $q$ in $S$.

- $Access(x)$ returns the symbol at the position $x$ of $S$.

Though there are a number of *Succinct Data Structures* available for real-world applications, we choose Wavelet Tree (Grossi et al. (2003) as the core of our method. We choose Wavelet Tree (WT) because there are already a number of evident successes in applying WT for large-scale, data-intensive applications, such as Data Store (Agarwal et al. (2015); Khandelwal et al. (2016)), Graph Processing (Khandelwal et al. (2017)) and etc. Therefore, our method deploys WT as the compression technique during the inference runtime.

### 4.1 Wavelet Tree

Wavelet Tree, a kind of *Succinct Data Structures* introduced by Grossi et al. (2003), was originally used in compressed suffix arrays. Since its initial use, a myriad of applications has been found. For instance, the wavelet tree could be used as representations of a sequence, a reordering of elements and grid of points (Navarro (2014)). We elaborate the details of WT as follows. Figure 2 gives an example of WT, where the alphabets/subsets are partitioned into pairs of subsets recursively, until the bit-vector can be distinguished by "0" and "1".

As a lossless compression method, WT is near-optimal. Assume a string S (length = n) is composed of $\sigma$ different symbols: if we use wavelet tree to represent this string, the space consumption is n log($\sigma$) + O(n) bits, which is near-optimal to the information-theoretic lower bound. Moreover, WT yields significant potentials for runtime performance. WT allows $Rank$, $Select$ and $Access$ operators to only take logarithmic time complexity. Following the above assumption: these operations are supported in O(log $\sigma$) time, where $\sigma$ is the size of the alphabet for the sequence. Note that the time complexity of these operators are independent to the string length, which can bring significant benefits to the computation without full decompression.

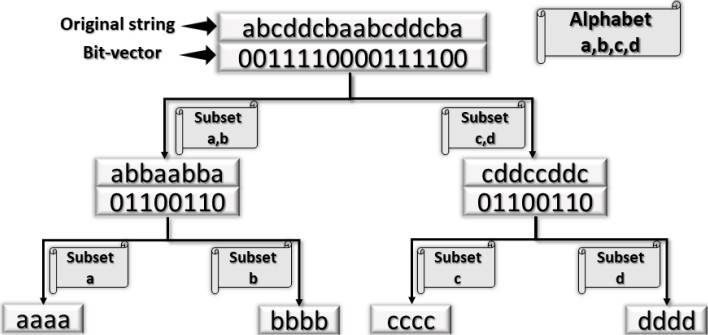

Figure 2: An example of Wavelet Tree (WT): This is an illustration of WT for the string "abcd-dcbaabcddcba". The bitvector for its first layer (i.e. 0011110000111100) is derived based on the alphabet partition: characters "a" and "b", located in the left subset of the alphabet, are encoded with the bit "0", and the other two characters "c" and "d" are encoded with the bit "1". For two subsets in the next layer, they are recursively divides into smaller subsets, which are "a" , "b", "c" and "d". Then the encoding is continued correspondingly. As a result, all characters can be distinguished by an unique sequence of bits.

## 5   MODEL INFERENCE IN *Succinct Data Structures*

The third stage of our method is to perform model inference via *Succinct Data Structures*. This stage is trivial to first retrieve all relevant operands via queries, then decode these operands, and finally perform the computation for model inference. Our methods can yield significant performance benefits because the queries on compressed models don't require any decompression. This is because carefully engineering $Rank$, $Select$ and $Access$ operators allow the queries directly on compression. We restrict each set of decompressed data for one convolution filter at a time. To extract these operands from *Succinct Data Structures*, we use two $Select$ operators to locate the corresponding values within the compressed representation; and then we use one $Access$ operator to retrieve the values from the range, restricted by the $Select$ operators. In practice, the inference process for Element-wise method slightly differs from that of Block-wise method: Element-wise method can perform the inference by extracting elementary operands individually from WT, and Block-wise method can only perform the inference by extracting aggregated sets of operands. This is because a block is the smallest unit for indexes in Block-wise method.

## 6   EXPERIMENTAL STUDY

We compare Succinct Compression with Huffman Coding(**?**), as suggested by Deep Compression (Han et al. (2016)), for four networks: AlexNet (Krizhevsky et al. (2012)), Pruned-Quantized AlexNet, VGG-16 Simonyan & Zisserman (2015) and Pruned-Quantized VGG-16 on ImageNet dataset (Deng et al. (2009)). For Pruning and Quantization, we perform the same method suggested by Deep Compression (Han et al. (2016)). Since Succinct Compression is lossless, we find that all models achieve the same level of accuracy, compared with model inference without Succinct Compression. We form an extension of Succinct Compression, by replacing *Succinct Data Structures* with Huffman Encoding as our baseline. All experiments were performed using PyTorch (Paszke et al. (2019)) on the Intel Core i7 5930K.

### 6.1   NEAR-OPTIMAL COMPRESSION DURING INFERENCE RUNTIME

We first examine the impacts of Succinct Compression with AlexNet and VGG-16 (w/out pruning and quantization). Table 1 and Table 2 show the comparison regarding the statistics of the original model and different compression schemes. We make two observations. First, our method achieves near-optimal compression rate among all models. Our methods has the best compression rate of 19.80% for AlexNet and 18.72% for VGG-16. This is credited to the proper applications of our method using *Succinct Data Structures*. Second, our method achieves significant reduction of layers

in terms of runtime space. Our method reduces AlexNet from 232.56 MB to 46.04 MB, and VGG-16 from 527.8 MB to 98.82 MB.

Table 1: Compression statistics for AlexNet (No PQ). CR: Compression Rate, HF: Huffman Coding, WT-E: Wavelet Tree via Element-Wise, WT-B: Wavelet Tree via Block-wise.

| Layer | Size (Origin) | Size (HF) | Size (WT-E) | Size (WT-B) | CR (HF) | CR (WT-E) | CR (WT-B) |
|---|---|---|---|---|---|---|---|
| conv1 | 0.1333 | 0.0359 | 0.0417 | 0.1239 | 26.89% | 31.32% | 92.92% |
| conv2 | 1.1729 | 0.2424 | 0.3310 | 0.4118 | 20.67% | 28.22% | 35.11% |
| conv3 | 3.3765 | 0.6443 | 0.9112 | 0.8559 | 19.08% | 26.99% | 25.35% |
| conv4 | 2.5327 | 0.4950 | 0.6827 | 0.6629 | 19.54% | 26.96% | 26.17% |
| conv5 | 1.6885 | 0.3339 | 0.4673 | 0.4741 | 19.77% | 27.68% | 28.08% |
| fc6 | 144.02 | 19.99 | 35.97 | 27.87 | 13.88% | 24.98% | 19.35% |
| fc7 | 64.02 | 9.649 | 17.99 | 12.47 | 15.07% | 28.10% | 19.48% |
| fc8 | 15.63 | 2.682 | 4.300 | 3.177 | 17.16% | 27.51% | 20.33% |
| conv_total | 8.904 | 1.751 | 2.434 | 2.529 | 19.67% | 27.34% | 28.40% |
| fc_total | 223.7 | 32.327 | 58.26 | 43.52 | 14.45% | 26.05% | 19.46% |
| all_total | 232.56 | 34.07 | 60.69 | 46.04 | 14.65% | 26.10% | 19.80% |

Table 2: Compression statistics for VGG-16 (No PQ). CR: Compression Rate, HF: Huffman Coding, WT-E: Wavelet Tree via Element-Wise, WT-B: Wavelet Tree via Block-wise.

| Layer | Size (Origin) | Size (HF) | Size (WT-E) | Size (WT-B) | CR (HF) | CR (WT-E) | CR (WT-B) |
|---|---|---|---|---|---|---|---|
| conv1_1 | 0.006834 | 0.00582 | 0.00499 | 0.007865 | 85.08% | 72.92% | 115.05% |
| conv1_2 | 0.1409 | 0.0345 | 0.04046 | 0.1159 | 24.48% | 28.72% | 82.25% |
| conv2_1 | 0.2817 | 0.0631 | 0.08107 | 0.1592 | 22.38% | 28.78% | 56.49% |
| conv2_2 | 0.5630 | 0.1160 | 0.1539 | 0.2271 | 20.61% | 27.33% | 40.34% |
| conv3_1 | 1.126 | 0.2153 | 0.3019 | 0.3335 | 19.12% | 26.81% | 29.62% |
| conv3_2 | 2.251 | 0.3991 | 0.5959 | 0.5380 | 17.72% | 26.47% | 23.90% |
| conv3_3 | 2.251 | 0.4001 | 0.5887 | 0.5404 | 17.77% | 26.15% | 24.01% |
| conv4_1 | 4.502 | 0.7578 | 1.155 | 0.9893 | 16.83% | 25.65% | 21.97% |
| conv4_2 | 9.002 | 1.411 | 2.202 | 1.843 | 15.68% | 24.46% | 20.47% |
| conv4_3 | 9.002 | 1.423 | 2.290 | 1.860 | 15.80% | 25.44% | 20.66% |
| conv5_1 | 9.002 | 1.466 | 2.437 | 1.877 | 16.29% | 27.08% | 20.85% |
| conv5_2 | 9.002 | 1.476 | 2.489 | 1.851 | 16.40% | 27.65% | 20.56% |
| conv5_3 | 9.002 | 1.458 | 2.501 | 1.841 | 16.19% | 27.78% | 20.45% |
| fc6 | 392.0 | 42.63 | 121.5 | 71.75 | 10.87% | 30.98% | 18.30% |
| fc7 | 64.02 | 8.642 | 19.41 | 11.89 | 13.50% | 30.33% | 18.58% |
| fc8 | 15.63 | 2.517 | 4.526 | 2.995 | 16.10% | 28.96% | 19.16% |
| conv_total | 56.13 | 9.226 | 14.84 | 12.18 | 16.44% | 26.44% | 21.70% |
| fc_total | 471.7 | 53.78 | 145.4 | 86.64 | 11.40% | 30.83% | 18.37% |
| all_total | 527.8 | 63.01 | 160.2 | 98.82 | 11.94% | 30.36% | 18.72% |

Interestingly, we also notice that Block-wise method outperforms Element-wise method in most cases. We find that, Block-wise method saves more space than Element-wise method for FC layers of AlexNet and all layers of VGG-16 in total. This is because Block-wise method can effectively reduce the overall size of the compressed representation, which refers to the required length for bit-encoding to distinguish different values, by using individual bits to represent more operands. However, we also observe that Element-wise method saves more space than Block-wise method requires for CONV layers of AlexNet. This is because CONV layers of AlexNet are relatively small, where the redundancy of Block-wise Method plays a significant role in space consumption.

We then examine the impacts of Pruning and Quantization on Succinct Compression. Table 3 and Table 4 show the comparison regarding the statistics of the pruned and quantized model and different compression schemes. We make two observations. First, we find that our method is very synergistic with Pruning and Quantization. For Pruned-Quantized AlexNet/VGG-16, our method achieves the compression rate with only 1% difference, compared with the optimal solution. Second, we find that Element-wise method achieves better space savings for both CONV and FC layers.

For CONV/FC layers, Element-wise method reduces runtime space consumption by 8%/0.5% on average, compared with Block-wise method.

Table 3: Compression statistics for AlexNet (w/ PQ). CR: Compression Rate, HF: Huffman Coding, WT-E: Wavelet Tree via Element-Wise, WT-B: Wavelet Tree via Block-wise.

| Layer | Size (Origin) | Size (HF) | Size (WT-E) | Size (WT-B) | CR (HF) | CR (WT-E) | CR (WT-B) |
|---|---|---|---|---|---|---|---|
| conv1 | 0.05589 | 0.0333 | 0.0371 | 0.04352 | 25.08% | 27.94% | 32.74% |
| conv2 | 0.2231 | 0.1576 | 0.1856 | 0.2116 | 13.44% | 15.84% | 18.06% |
| conv3 | 0.5912 | 0.4081 | 0.5473 | 0.8420 | 12.09% | 16.22% | 24.95% |
| conv4 | 0.4681 | 0.3230 | 0.4304 | 0.6408 | 12.76% | 17.00% | 25.31% |
| conv5 | 0.3119 | 0.2180 | 0.2888 | 0.4268 | 12.92% | 17.11% | 25.29% |
| fc6 | 8.325 | 4.002 | 4.685 | 5.494 | 2.779% | 3.253% | 3.815% |
| fc7 | 3.697 | 1.778 | 2.083 | 2.448 | 2.777% | 3.255% | 3.825% |
| fc8 | 1.972 | 1.011 | 1.113 | 1.121 | 6.473% | 7.122% | 7.176% |
| conv_total | 1.650 | 1.140 | 1.489 | 2.165 | 12.80% | 16.73% | 24.31% |
| fc_total | 13.99 | 6.791 | 7.881 | 9.063 | 3.04% | 3.52% | 4.05% |
| all_total | 15.64 | 7.931 | 9.370 | 11.23 | 3.41% | 4.03% | 4.83% |

Table 4: Compression statistics for VGG-16 (w/ PQ). CR: Compression Rate, HF: Huffman Coding, WT-E: Wavelet Tree via Element-Wise, WT-B: Wavelet Tree via Block-wise.

| Layer | Size (Origin) | Size (HF) | Size (WT-E) | Size (WT-B) | CR (HF) | CR (WT-E) | CR (WT-B) |
|---|---|---|---|---|---|---|---|
| conv1_1 | 0.001977 | 0.002652 | 0.003235 | 0.002836 | 40.23% | 49.08% | 43.02% |
| conv1_2 | 0.01528 | 0.01983 | 0.02061 | 0.02423 | 14.10% | 14.66% | 17.23% |
| conv2_1 | 0.04790 | 0.04081 | 0.04824 | 0.06328 | 14.51% | 17.15% | 22.50% |
| conv2_2 | 0.1006 | 0.07720 | 0.09635 | 0.1342 | 13.73% | 17.13% | 23.85% |
| conv3_1 | 0.2974 | 0.1920 | 0.2510 | 0.3496 | 17.07% | 22.31% | 31.07% |
| conv3_2 | 0.2700 | 0.2090 | 0.2810 | 0.4303 | 9.290% | 12.49% | 19.13% |
| conv3_3 | 0.4725 | 0.3177 | 0.4210 | 0.6171 | 14.12% | 18.71% | 27.43% |
| conv4_1 | 0.7186 | 0.5048 | 0.6829 | 1.0641 | 11.22% | 15.18% | 23.65% |
| conv4_2 | 1.214 | 0.8676 | 1.234 | 1.959 | 9.640% | 13.72% | 21.77% |
| conv4_3 | 1.528 | 1.046 | 1.454 | 2.266 | 11.62% | 16.16% | 25.17% |
| conv5_1 | 1.576 | 1.071 | 1.485 | 2.306 | 11.90% | 16.49% | 25.62% |
| conv5_2 | 1.306 | 0.9176 | 1.266 | 2.050 | 10.20% | 14.06% | 22.78% |
| conv5_3 | 1.622 | 1.095 | 1.485 | 2.350 | 12.17% | 16.50% | 26.11% |
| fc6 | 10.75 | 5.430 | 6.726 | 7.817 | 1.385% | 1.716% | 1.994% |
| fc7 | 1.755 | 0.8897 | 1.100 | 1.283 | 1.390% | 1.719% | 2.004% |
| fc8 | 1.795 | 0.9485 | 1.128 | 1.125 | 6.071% | 7.219% | 7.199% |
| conv_total | 9.171 | 6.361 | 8.728 | 13.62 | 11.33% | 15.55% | 24.26% |
| fc_total | 14.30 | 7.268 | 8.954 | 10.22 | 1.54% | 1.90% | 2.17% |
| all_total | 23.47 | 13.63 | 17.68 | 23.84 | 2.58% | 3.35% | 4.52% |

Note that our observation is slightly different from the results from models without PQ, where the results suggest that Block-wise saves more space for FC layers. This is because Pruning and Quantization significantly reduces the number of parameters in FC layers, which made the redundancy of Block-wise method becomes the major limiter for compression.

## 6.2 SPEEDUP-COMPRESSION RATE TRADEOFF

We report the Speedup-Compression Rate Tradeoff in Figure 3 for all models. We also break down the results as all layers, CONV layers and FC layers separately. We find that our method achieves at least 8.65X speedup for the inference of all layers. And for CONV/FC layers, our method achieves speedup 10.47X/1.6X at least, except for the FC layers in VGG-16 without Pruning and Quantization. The exception is because Block-wise method incurs significant redundancy, since both the alphabet of WT and the amount of parameters are huge and grow significantly.

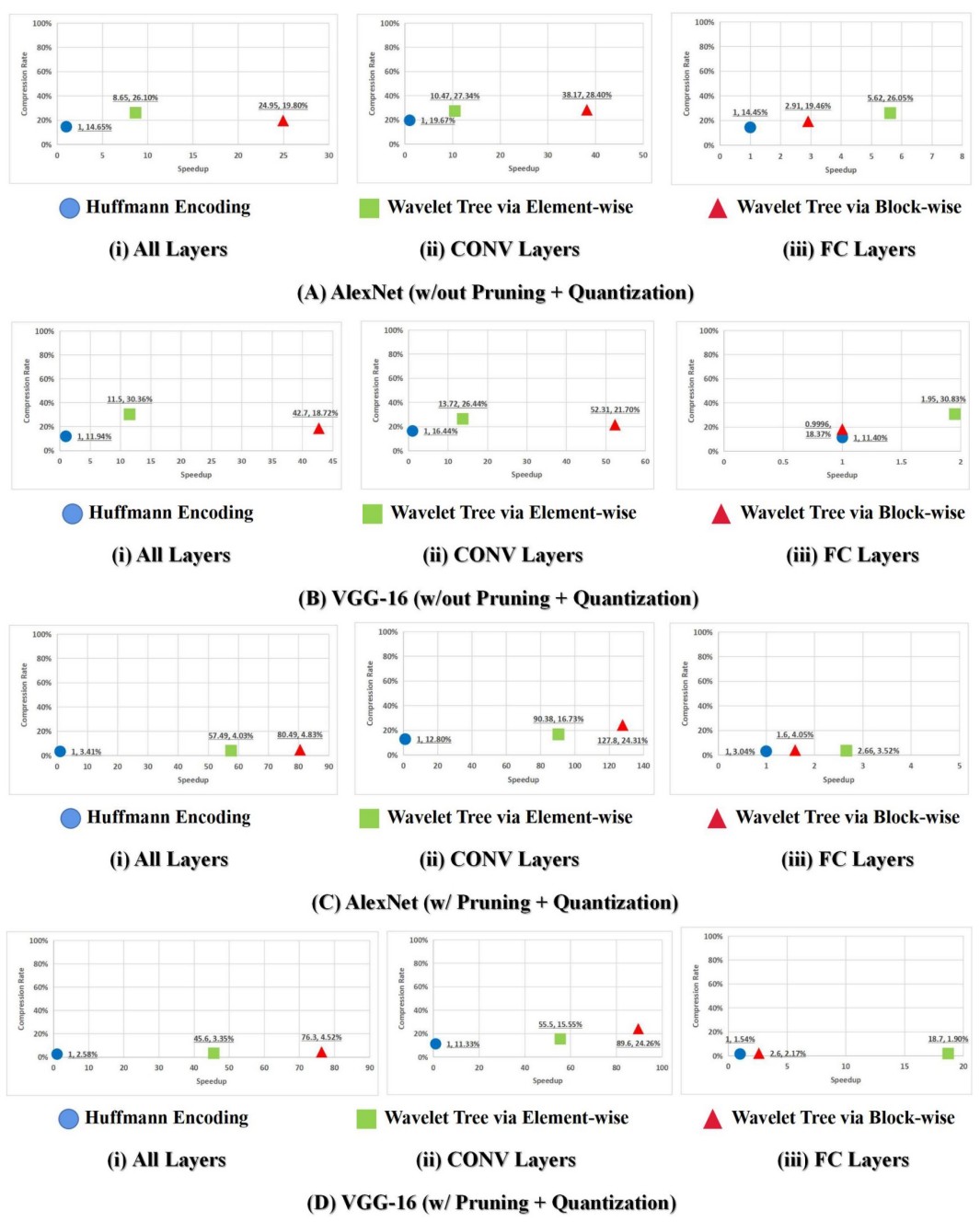

Figure 3: Speedup-Compression Rate Tradeoff of Huffman Coding, our Element-wise method, and our Block-wise method among (A) AlexNet (w/out PQ); (B) VGG-16 (w/out PQ); (C) AlexNet (w/ PQ); and (D) VGG-16 (w/ PQ). For each sub-figure, results are for (i) all layers; (ii) CONV layers; (iii) FC layers. The y-axis refers to Compression Rate (lower is better), and the x-axis refers to Speedup (higher is better).

Interestingly, for FC layers, we observe variations across compression schemes and models. We make two observations. First, Element-wise method achieves better speedup with worse compression rate for FC layers of models w/out PQ. This is because Element-wise formulation suits better for the structure of FC layers, but also suffers from a huge amount of parameters. Second, Element-wise method achieves both better speedup and compression rate for FC layers of models w/ PQ.

This is because PQ can significantly reduces the amount of parameters, which make Element-wise method more suitable.

### 6.3 SPEEDUP-LAYER SIZE TRADEOFF

We finally examine the tradeoff between the speedup and layer size among different methods. Note that, after PQ, the permutation of layers from AlexNet remain consistent, but VGG-16 changes slightly in CONV layers. We draw two observations. First, the benefits of Element-wise method exhibits more robust trends in CONV layers in most cases, with increasing speedup with the growing layer size, compared with Block-wise method. This is because, compared with Element-wise method, Block-wise method is much more sensitive to both layer size and the layer structure. Second, PQ significantly varies the tradeoffs between speedup and layer size. This is because PQ exploits the structure information to reduce the layer size as well as the layer complexity. Therefore, it's expected that the speedup may vary significantly due to PQ.

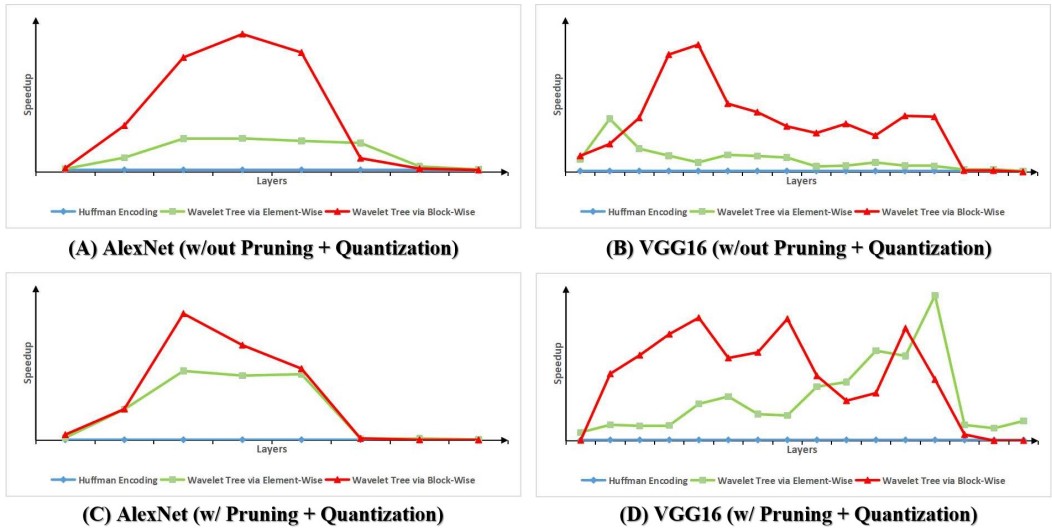

Figure 4: Speedup-Layer Size tradeoffs among (A) AlexNet (w/out PQ); (B) VGG-16 (w/out PQ); (C) AlexNet (w/ PQ); and (D) VGG-16 (w/ PQ).

## 7 DISCUSSIONS

Though we demonstrate the synergy between our method and Pruning-Quantization from Han et al. (2016), different Pruning and Quantization methods are expected to be synergistic as well. This is because our method doesn't affect any optimization in terms of model structures and value encoding, as Pruning and Quantization achieves. In addition, our element-wise and block-wise methods can be combined for a single model inference, since it can be constructed in a layer-wise manner. As suggested by our results, the layer heterogeneity breeds the needs to combine different formulations to obtain the best tradeoffs between performance and space consumption.

## 8 CONCLUSIONS

We present "Succinct Compression" that improves inference performance with compressed neural networks during inference runtime. Our method operates by proper combinations of newly-introduced formulations, *Succinct Data Structures* and carefully-engineered inference. We show our method achieves near-optimal compression results, compared with Huffman Coding. In addition, we reveals that our method achieves significant speedup over Huffman Coding, by exploiting the characteristics of queries directly on compression in *Succinct Data Structures*. We also show that our method is synergistic with Pruning and Quantization, which brings significant performance benefits and space saving at the same time.

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
