# OpenReview forum: "Succinct Compression: Near-Optimal and Lossless Compression of Deep Neural Networks during Inference Runtime"
_ICLR.cc/2022/Conference — ICLR 2022 Submitted_

### Official Review · Reviewer_QdUC · 2021-10-30

**Correctness:** 3
**Technical Novelty And Significance:** 2
**Empirical Novelty And Significance:** 2
**Recommendation:** 3
**Confidence:** 4

**Main Review:**

The paper discusses an interesting and important problem and proposes what appears to be a novel approach. There are two main issues for me at the moment 1) it is difficult to understand the precise scheme that has been proposed, and 2) the evaluation seems very limited and somewhat unclear.

1. To confirm, pruning and quantization was performed but only to the point that the model accuracy did not change?
2. What were the results of quantization?
3. When pruning/quantization is applied, wouldn't it be clearer to use 15.64MB as the new baseline and report CR compared to this. Claiming "only 1% difference" hides the fact that the HF model size = 6.8 and with WT-B it is 9.063. Obviously, the overall CR is low if you compare to the original model size of 232MB.
4. Again, results for the performance of the quantized and quantized/pruned models would be of interest, but I don't think are presented? (i.e. without your compression scheme). There are many simple data formats suitable for stored sparse data of course (e.g. using sparsity maps or Z-RLC, CSC/CSR perhaps). Would these be faster but use slightly larger models?
5. You don't compare to other simple compression schemes, e.g. Dynamic Prediction Reduction (DPRed) perhaps?
6. Are there variations of the Wavelet-tree scheme that could have been evaluated?
7. I don't think you can say Huffman encoding is strictly optimal?
8. Speedups are achieved through the reduction of memory traffic I assume, what is the overhead of accessing these compressed formats? e.g. if applied to a small model that fitted in the cache? I assume the architecture of the target processor has a big impact here too and the overhead will perhaps be smallest on the desktop style processor you target. How would the results differ on embedded cores for instance?

Low-rank approximation would be another technique to discuss or compare to perhaps.

I found some parts of the paper to be difficult to follow unfortunately, e.g. page 3. I suspect this is quite simple, but I struggled. Perhaps better diagrams or an example would help.

The bitmap Figures are also very difficult to see (e.g. Figure 1 and Figure 3 - the y-axis could perhaps cover a smaller range, avoding bitmaps would help too.

I understand the technique is lossless, but it would perhaps still be useful to confirm the model accuracy.

minor:
* "aimming"
* missing reference for Huffman "(?)"


**Summary Of The Paper:**

The paper explores the compression of weights in a DNN. The compression technique presented permits the uncompressed values to be easily recovered without first decompressing the data. The approach appears to take advantage of Wavelet Trees after the data has been structured in some way.

Results are presented showing compression rates (vs. Huffman encoding) and speedups on a CPU.

**Summary Of The Review:**

I feel that the description of the work needs to be improved. The evaluation also needs improvement. For these reasons I am recommending the paper is rejected.

---

> ### Author Response · Authors · 2021-11-17
> **Response to reviewer QdUC**
>
> We thank the reviewer QdUC for the constructive and valuable feedback.
>
> Q1: To confirm, pruning and quantization was performed but only to the point that the model accuracy did not change?
>
> A1: Yes, we just perform PQ to the extent that doesn’t influence model accuracy to make a comparison with Han’s Deep Compression. More space-efficient PQ could be conduct, but this might be against our claim: “lossless compression”.
>
> Q2: What were the results of quantization?
>
> A3: You could see part of the quantization results in figure 1. As shown in the figure, the second element in the tuple like (1, 1.01) is the quantization result. This is slightly different from that of conventional quantization result, since there is no need to replace floating point number with its corresponding category index. In the context of succinct compression, using category index will consume more space and reduce the query efficiency. Due to this, we finally choose to use the floating-point number directly.
>
> Q3: When pruning/quantization is applied, wouldn't it be clearer to use 15.64MB as the new baseline and report CR compared to this. Claiming "only 1% difference" hides the fact that the HF model size = 6.8 and with WT-B it is 9.063. Obviously, the overall CR is low if you compare to the original model size of 232MB.
>
> A3: We will fix this in a later version.
>
> Q4: Again, results for the performance of the quantized and quantized/pruned models would be of interest, but I don't think are presented? (i.e. without your compression scheme). There are many simple data formats suitable for stored sparse data of course (e.g. using sparsity maps or Z-RLC, CSC/CSR perhaps). Would these be faster but use slightly larger models?
>
> A4: We would consider this in a later version.
>
> Q5: You don't compare to other simple compression schemes, e.g. Dynamic Prediction Reduction (DPRed) perhaps?
>
> A5: We will add a section to differentiate our method with them.
>
> Q6: Are there variations of the Wavelet-tree scheme that could have been evaluated?
>
> A6: We will add a section to evaluate variants of Wavelet tree.
>
> Q7: I don't think you can say Huffman encoding is strictly optimal?
>
> A7: Huffman encoding is indeed not strictly optimal. We apologize for the confusing writing.
>
> Q8: Speedups are achieved through the reduction of memory traffic I assume, what is the overhead of accessing these compressed formats? e.g. if applied to a small model that fitted in the cache? I assume the architecture of the target processor has a big impact here too and the overhead will perhaps be smallest on the desktop style processor you target. How would the results differ on embedded cores for instance?
>
> A8: We will run our experiment on different processors.
>
> Q9: Low-rank approximation would be another technique to discuss or compare to perhaps.
>
> A9: We will discuss this in a later version.
>
> Q10: (1)I found some parts of the paper to be difficult to follow unfortunately, e.g. page 3. I suspect this is quite simple, but I struggled. Perhaps better diagrams or an example would help. (2)The bitmap Figures are also very difficult to see (e.g. Figure 1 and Figure 3 - the y-axis could perhaps cover a smaller range, avoding bitmaps would help too.
>
> A10: We will fix these problems.
>
> Q11: I understand the technique is lossless, but it would perhaps still be useful to confirm the model accuracy.
>
> A11: We will show the accuracy in the next version.
>
> Q12: Minor: (1)”aimming” (2)missing reference
>
> A12: We will fix these problems.

---

> > ### Comment · Reviewer_QdUC · 2021-12-05
> > **Feedback after authors' response**
> >
> > I appreciate the clear responses from the authors to my questions. In this case I have decided to confirm my score.

---

### Official Review · Reviewer_R8Un · 2021-10-31

**Correctness:** 2
**Technical Novelty And Significance:** 3
**Empirical Novelty And Significance:** 3
**Recommendation:** 3
**Confidence:** 3

**Main Review:**

Paper strengths:

+ The idea of proposing a different data structure to store a neural network is novel for accelerating inference.

+ The paper shows good performance in the experimental part compared to the Huffman encoding.


Paper weaknesses:

- Claim on decompress compressed model requirement: There is a plethora of prior work that does not require the model to be decompressed during inference. For instance, distillation approaches, as well as factorization and pruning methods. The paper is motivated by a claim, which does not hold based on the prior work. This is also reflected in the experimental section which prior works from the aforementioned categories are completely ignored in the comparisons with the prior work. The claims of the work are a major limitation.

- The proposed data structure for the neural network is method agnostic. It would help the paper to perform experiments based on several quantizations and pruning approaches. This experiment would add a significant value to the work.

- The main contribution is how to formulate the compressed neural network (in terms of data structure) to accelerate the inference. This idea is not quite clear after reading the paper. Overall, the writing part needs some improvement to improve the clarity of the work.

- Related work on lossless compression: It is important to extensively discuss the related work on lossless compression and memory-efficient deployment. A few useful references:
	- Lossless Compression of Structured Convolutional Models via Lifting (2021).
	- Lossless Compression of Deep Neural Networks (2020).
	- Universal deep neural network compression (2020).
	- Hardware-Based Real-Time Deep Neural Network Lossless Weights Compression (2020)
	- Compact and Computationally Efficient Representation of Deep Neural Networks (2019).

- Prior work comparisons: There is only one comparison with the Huffman encoding. It would be helpful to include more approaches (see above).

- It would be helpful to include a ResNet-like architecture since it is the standard to be used nowadays.


Improvements:

- The claims in the introduction could be supported by references.

- "As revealed by prior works, the impacts of these compression schemes.." references are missing.

- The quality of the figures, e.g. Fig 1/2/3, is poor. Rendering on a better resolution is recommended.

**Summary Of The Paper:**

It is presented an approach to store a compressed deep neural network in a form that allows fast inference. The deep neural network is defined as an element- and block-wise Runtime Accessible Sequence (RAS). Moreover, the RAS elementary operands are defined as tuples to account for parameter quantization and pruning. Given the neural network RAS formulation, succinct data structures, i.e. wavelet tree, are used to store the network during inference.  The approach is evaluated on AlexNet and VGG-16 architectures which are quantized and pruned. The results are promising compared to the Huffman encoding.

**Summary Of The Review:**

Overall, the paper presents an interesting idea. It is relatively easy to follow it, but the clarity of the paper needs improvement. The experimental part needs additional work, as already discussed. There are major claims which might need to be reformulated. There are also missing references at different points. Overall, the work is not ready for publication yet.

---

> ### Author Response · Authors · 2021-11-17
> **Response to reviewer R8Un**
>
> We thank the reviewer R8Un for the constructive and valuable feedback.
>
> Q1: The paper is motivated by a claim, which does not hold based on the prior work. This is also reflected in the experimental section which prior works from the aforementioned categories are completely ignored in the comparisons with the prior work. The claims of the work are a major limitation.
>
> A1: We will fix the claim.
>
> Q2: The proposed data structure for the neural network is method agnostic. It would help the paper to perform experiments based on several quantizations and pruning approaches. This experiment would add a significant value to the work.
>
> A2: Thanks for your thoughtful advice. We will verify our method’s compatibility with other quantization and pruning approaches in a later version.
>
> Q3: How to formulate the compressed neural network (in terms of data structure) is not quite clear after reading the paper.
>
> A3: We will address this issue in a later version.
>
> Q4: Related work on lossless compression: It is important to extensively discuss the related work on lossless compression and memory-efficient deployment.
>
> A4: We will discuss this in the later version.
>
> Q5: Prior work comparisons: There is only one comparison with the Huffman encoding. It would be helpful to include more approaches.
>
> A5: We will add more comparisons.
>
> Q6: It would be helpful to include a ResNet-like architecture since it is the standard to be used nowadays.
>
> A6: We will conduct experiment on a ResNet-like architecture.
>
> Q7: Improvements: (1) The claims in the introduction could be supported by references. (2) "As revealed by prior works, the impacts of these compression schemes.." references are missing. (3) The quality of the figures, e.g. Fig 1/2/3, is poor. Rendering on a better resolution is recommended.
>
> A7: We will fix these problems.

---

> > ### Comment · Reviewer_R8Un · 2021-11-18
> > **Feedback after reading the reviews and authors response**
> >
> > The authors took into consideration all comments of the reviews. Nevertheless, the are signficant changes to be made. The work needs to be further improved and submitted again in future.

---

### Official Review · Reviewer_G6tm · 2021-11-02

**Correctness:** 3
**Technical Novelty And Significance:** 2
**Empirical Novelty And Significance:** 2
**Recommendation:** 3
**Confidence:** 4

**Main Review:**

Strengths:

- The paper explores an interesting idea to perform lossless compression for efficient DNN inference.
- The proposed method seems to be able to achieve a better compression rate than Huffman Coding, a compression technique used in DeepCompression from Han et al. 2015.

Weaknesses:
- The missing discussion of related work makes the significance of the contribution questionable.
- The comparison with existing work is weak.
- The paper lacks details on how speedups are measured, which is important for model inferencing.
- Writing needs to be improved.


**Summary Of The Paper:**

The paper introduces a method to compress DNN networks with succinct data structures for efficient inference. The paper claims that DNN inference suffers from a storage bottleneck and proposes a lossless compression scheme using wavelet trees to reduce the size of DNN models. The paper performs an evaluation of the proposed method on AlexNet/VGG-16 and shows that the method outperforms existing methods for compressing models using Huffman Coding.

**Summary Of The Review:**

The paper explores an interesting idea for compressing DNN networks, e.g., through lossless compression. This could be potentially quite impactable but the current state of the paper raises several concerns:

1. There have been significant missing discussions related to model compression. In particular, if we look that the related work section, it is very weird that most of the cited papers in Section 2 are work done prior to 2017, whereas there have been vast advancements for model compression in the past several years that the paper seems to choose to ignore, such as [1-3]. Of course, many existing compression techniques are not lossless in theory, but they often achieve comparable performance as the original model, e.g., quantization, that cannot be dismissed completely.

2. The evaluation section is weak. The comparison is made primarily with the Huffman coding method used in Deep Compression [4], which was done six years ago and no longer represents the state-of-the-art. Furthermore, the evaluation was also done on AlexNet and VGG, both of which have been heavily studied. To be more convincing, the paper should compare at least with some lossy compression techniques such as integer quantization and one more advanced model architecture such as ResNet and Transformers.

3. The paper lacks descriptions of implementations and how speedups are calculated. Is the speedup measured through proxies or measured execution time? Is Succinct Data structure GPU hardware friendly? How does it perform on GPU in comparison with models served via cuDNN?  What are the implementations used for the baseline?

4. The paper would benefit from thorough proofreading.

Typos:
1. Missing references, e.g. Huffman Coding (?), in the first paragraph of Section 6.

[1] Dong et al. HAWQ: Hessian AWare Quantization of Neural Networks with Mixed-Precision, https://arxiv.org/abs/1905.03696, 2019

[2] Choudhary et al. A comprehensive survey on model compression and acceleration, 2020

[3] Deng et al. A comprehensive survey on model compression and acceleration, 2020

[4] Cho et al. On the Efficacy of Knowledge Distillation, 2019

[5] Deep Compression: Compressing Deep Neural Networks with Pruning, Trained Quantization and Huffman Coding, Han et al., 2015, https://arxiv.org/abs/1510.00149

---

> ### Author Response · Authors · 2021-11-17
> **Response to reviewer G6tm**
>
> We thank the reviewer G6tm for the constructive and valuable feedback.
>
> Q1: There have been significant missing discussions related to model compression.
>
> A1: We will revise this.
>
> Q2: To be more convincing, the paper should compare at least with some lossy compression techniques such as integer quantization and one more advanced model architecture such as ResNet and Transformers.
>
> A2: We will do more comparisons.
>
> Q3: The paper lacks descriptions of implementations and how speedups are calculated.
>
> A3: The speedup is (Huffman encoding’s execution time) / (Succinct compression’s execution time).
>
> Q4: Typos: missing reference
>
> A4: We will fix this.

---

> > ### Comment · Reviewer_G6tm · 2021-12-05
> > **Reply to the authors**
> >
> > Thanks for responding to the reviews, and the reviewer would like to keep the score. The reviewer also encourages the authors to take into the feedback to improve the paper.

---

### Official Review · Reviewer_eHWE · 2021-11-03

**Correctness:** 2
**Technical Novelty And Significance:** 2
**Empirical Novelty And Significance:** 2
**Recommendation:** 3
**Confidence:** 3

**Main Review:**

The idea of enabling inference directly on the compressed model without decompression is quite interesting. However, I could not verify if the proposed method indeed shows any improvement in runtime speedup or compression rate, or accuracy by looking at the experimental results. It is also not very straightforward to me that there must be a speedup without any empirical support because all the weights need to be queried for inference and I do not see how this is different from decompression. Here are my main concerns and questions:

- The paper is not very easy to follow and needs significant improvement in writing. There are repeating sentences, many typos, and the general flow is not very fluent. I especially recommend making Section 3 more clear since it is the most important section of the paper and some concepts need to be better explained by providing some background information.

- I believe the related work section misses many references, especially in the model encoding part, which is the most related direction to this paper. For example, [1], [2], [3], and [4] would be good candidates for the related work section since they also propose new methods to encode neural network models.

- The authors compare their work with Huffman coding in Deep Compression paper [5]. However, there are more recent methods such as [1], [2], [3], and [4] that are directly comparable. A comparison with these papers would strengthen the paper.

- The authors claim that their method is near-optimal but they do not provide any reason as to why this is the case. In Section 4.1, they mention that the space consumption is $n \log(\sigma) + O(n)$ bits and claim that this is near-optimal to the information-theoretic lower bound. A $n \log(\sigma)$ space consumption is actually an upper bound, which is exact only when the weights are uniformly distributed. There are several empirical results that show that weights are not uniformly distributed, e.g., [2] shows that trained weights look Laplacian. Therefore, the proposed method has a space consumption close to the worst-case estimate of the entropy, which is not close to the optimal model compression since we know that weights are not uniform.

- $\sigma$ is used for both the alphabet and the alphabet size in different places. It would be nice to fix this to avoid confusion.

- There are some missing citations. For example Huffman Coding(?) in page 5.

- Maybe most importantly, the authors claim in Section 6.1 that the proposed method has the best compression rate of $19.8$% for AlexNet and $18.72$% for VGG-16.  However, by looking at Tables 1 and 2, the only baseline (Huffman Coding) has a better compression rate of $14.65$% and $11.94$%, respectively. It means that the text and the results in the tables are not consistent, and the proposed method underperforms Huffman Coding.

- I find the following sentence from the paper very confusing: "For Pruned-Quantized AlexNet/VGG-16, our method
achieves the compression rate with only $1 \%$ difference, compared with the optimal solution.". Can the authors explain what the optimal solution here is?


[1] Wiedemann, Simon, et al. "Deepcabac: Context-adaptive binary arithmetic coding for deep neural network compression." arXiv preprint arXiv:1905.08318 (2019).

[2] **v1 of** Isik, Berivan, Albert No, and Tsachy Weissman. "Rate-Distortion Theoretic Model Compression: Successive Refinement for Pruning." arXiv preprint arXiv:2102.08329 (2021).

[3] Havasi, Marton, Robert Peharz, and José Miguel Hernández-Lobato. "Minimal random code learning: Getting bits back from compressed model parameters." arXiv preprint arXiv:1810.00440 (2018).

[4] Oktay, Deniz, et al. "Scalable model compression by entropy penalized reparameterization." arXiv preprint arXiv:1906.06624 (2019).

[5] Han, Song, Huizi Mao, and William J. Dally. "Deep compression: Compressing deep neural networks with pruning, trained quantization and huffman coding." arXiv preprint arXiv:1510.00149 (2015).

**Summary Of The Paper:**

This paper aims to improve the runtime of NN inference via a novel DNN compression idea. Specifically, the authors propose an approach that allows fast queries at inference directly on the compressed model without decompressing the model back. They show comparisons to prior work on AlexNet andVGG16 architectures.

**Summary Of The Review:**

Overall, I think the paper is not ready for publication due to the reasons I listed in the previous section.

---

> ### Author Response · Authors · 2021-11-17
> **Response to reviewer eHWE**
>
> We thank the reviewer eHWE for the constructive and valuable feedback.
>
> Q1: I could not verify if the proposed method indeed shows any improvement in runtime speedup or compression rate, or accuracy by looking at the experimental results.
>
> A1: (1) Our speedup is the result of well-exploited data locality and efficient query strategy. The extremely small model size makes it suitable to place model in caches, which significantly reduces the access time. Meanwhile, the query strategy just brings about negligible overhead. Owe to these two features, our method achieves runtime speedup.
>
> (2) Compared to pruned/quantization models, our method has an improvement in terms of compression rate. As shown in Han’s Deep Compression, improvement on compression rate is still possible even after pruning and quantization, since the encoding of the model is not optimal. Based on this observation, we introduce a new encoding approach, Wavelet tree, and achieve compression improvement.
>
> (3) Succinct data structures compress data in a lossless way, so the accuracy of the models will not change.
>
>
> Q2: It is also not very straightforward to me that there must be a speedup without any empirical support because all the weights need to be queried for inference and I do not see how this is different from decompression.
>
> A2: Speedup over conventional compression methods come from the well-exploited data locality. With a strict limitation on runtime memory resources (e.g., Intel i7-4700EC CPU caches have only 8MB), our method can significantly outperform Huffman coding because we only need one-time traversal of all compressed data, while the Huffman-coded models must be decompressed for several times, to keep the runtime memory occupation within the limited boundary.
>
>
> Q3: I especially recommend making Section 3 more clear.
>
> A3: We will revise them.
>
>
> Q4: I believe the related work section misses many references, especially in the model encoding part.
>
> A4: We will revise them.
>
>
> Q5: The authors compare their work with Huffman coding in Deep Compression paper. However, there are more recent methods such as [1], [2], [3], and [4] that are directly comparable.
>
> A5: We will add comparisons with them in later versions.
>
>
> Q6: The authors claim that their method is near-optimal but they do not provide any reason as to why this is the case.
>
> A6: Our method can’t reach the optimal compression effect. However, compared to other compression methods like Huffman coding which take the uniform distribution of the parameters into consideration, our method doesn’t lag them a lot in terms of compression rate. The initial intention of using the word “near-optimal” is just to show our promising compression effect among various compression methods. Hereby, we apology for our unconcise writing which causes confusion.
>
>
> Q7: σ is used for both the alphabet and the alphabet size in different places. It would be nice to fix this to avoid confusion.
>
> A7: We will fix this.
>
>
> Q8: There are some missing citations.
>
> A8: We will revise this.
>
>
> Q9: In section 6.1, the text and the results in the tables are not consistent, and the proposed method underperforms Huffman Coding.
>
> A9: The phrase “best compression rate”, we use here, doesn’t mean the best result among HF, WT-E and WT-B but refers to the best compression effect occurring when we apply WT-E or WT-B. Hereby, we apologize for our unconcise writing which causes confusion.
>
>
> Q10: "For Pruned-Quantized AlexNet/VGG-16, our method achieves the compression rate with only 1 difference, compared with the optimal solution.". Can the authors explain what the optimal solution here is?
>
> A10: The optimal solution here is Huffman coding.

---

> > ### Comment · Reviewer_eHWE · 2021-12-02
> > **Response to Rebuttal**
> >
> > I thank the authors for their response. In my opinion, the authors have a nice idea but their claims currently do not have enough empirical support. For instance, the authors say that their method brings a runtime speedup in the paper (and in the rebuttal). But it is not clear how much speedup they provide compared to existing methods. The authors propose an alternative to Huffman coding for encoding models but according to the paper (and also the author feedback), their method is not doing as well as Huffman coding in terms of compression rate. I agree that it is not necessary to reach the performance of Huffman coding if their method brings other benefits such as runtime speedup as the authors claim. But in that case, I think it is necessary to provide more details and numerical results to show this speedup. Overall, I think the authors have a promising idea but the paper needs some work, especially in the writing and experiments section.

---

### Decision · Program_Chairs · 2022-01-20

**Decision:**

Reject

**Comment:**

### Summary

The paper proposes a technique that enables inference directly on a compressed model without decompressing the model.

### Discussion

- Strengths
  - An important problem as well as a compelling direction, namely inference without decompression.

- Weaknesses:
  - The reviewers provided a number of both broad and specific criticisms of the work.
  The most salient point is the lack of comparison to modern baselines.  Notably, the primary comparison is to a 2015 technique that, while seminal, has since been followed by significant related work (e.g, that identified by Reviewer eHWE, R8Un, and G6tm). In concert, the evaluation should consider at least one more contemporary network in the domain, such as a ResNet.

### Recommendation

I recommend Reject.  At current, this work is the first step in a strong, compelling direction. However, the work needs to be contextualized within a more modern context of contemporary results